# Sparsifying networks by traversing Geodesics

## Abstract

The geometry of weight spaces and functional manifolds of neural networks play
an important role towards 'understanding' the intricacies of ML. In this paper,
we attempt to solve certain open questions in ML, by viewing them through the
lens of geometry, ultimately relating it to the discovery of points or paths of
equivalent function in these spaces. We propose a mathematical framework to
evaluate geodesics in the functional space, to find high-performance paths from
a dense network to its sparser counterpart. Our results are obtained on VGG-11
trained on CIFAR-10 and MLP's trained on MNIST. Broadly, we demonstrate that
the framework is general, and can be applied to a wide variety of problems, ranging
from sparsification to alleviating catastrophic forgetting.

## 1 Introduction

The geometry of weight manifolds and functional spaces represented by artificial neural networks is
an important window to 'understanding' machine learning. Many open questions in machine learning,
when viewed through the lens of geometry, can be related to finding points or paths of equivalent
function in these spaces.

Two prominent examples are (i) Enabling networks to learn multiple tasks sequentially while avoiding
catastrophic forgetting[1] and (ii) to discover high-performance paths from a dense neural network to
its sparser counterparts (sparsification) [2, 3]. Both these questions, although appearing very different,
can be adequately solved by formulating paths in the weight manifold.

In this paper, we propose a mathematical framework grounded in differential geometry to find novel
solutions to these problems in machine learning. We formalize the "search" for a suitable network
as a dynamic movement on a curved pseudo-Riemannian manifold [4]. Further, we demonstrate
that geodesics, minimum length paths, on the weight manifold provide high performance paths that
the network can traverse to maintain performance while 'searching-out' for networks that satisfy
additional objectives. Specifically we develop a procedure based on the geodesic equation for
finding sets of path connected networks that achieve high performance while also satisfying a second
objective like sparsification. Broadly, our work provides procedures that will help discover an array
of high-performing neural network architectures for the task of interest.

## 2 Mathematical framework

**Preliminaries**: We represent a feed-forward neural network as a smooth, $\mathbb{C}^{\infty}$ function $f(\mathbf{x}, \mathbf{w})$, that
maps an input vector, $\mathbf{x} \in \mathbb{R}^k$, to an output vector, $f(\mathbf{x}, \mathbf{w}) = \mathbf{y} \in \mathbb{R}^m$. The function, $f(\mathbf{x}, \mathbf{w})$, is
parameterized by a vector of weights, $\mathbf{w} \in \mathbb{R}^n$, that are typically set in training to solve a specific
task. We refer to $W = \mathbb{R}^n$ as the *weight space* ($W$) of the network, and we refer to $\mathcal{F} = \mathbb{R}^m$ as
the *functional manifold* [5]. In addition to $f$, we will sometimes be interested in considering a loss

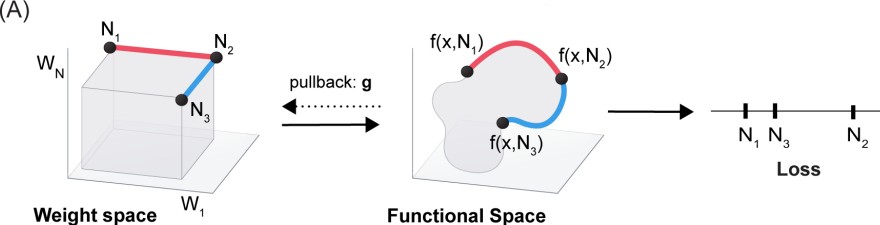

Figure 1: **Geometric framework for analyzing neural network resilience** (A) Three networks $(N_1, N_2, N_3)$ in weights space $W$ and their relative distance in functional space and loss space. Damage is analyzed by asking how movement in weight space changes functional performance and loss through introduction of a pullback metric **g**.

function, $L : \mathbb{R}^m \times \mathbb{R} \to \mathbb{R}$, that provides a scalar measure of network performance for a given task (Figure 1).

**Construction of metric tensor (g):** We use differential geometry, rooted in a functional notion of distance, to analyse how infinitesimal movements in the weight space ($W$) impact functional performance of the network. Specifically, we construct a local distance metric, **g**, that can be applied at any point in $W$ to measure the functional impact of an arbitrary network perturbation.

To construct a metric mathematically, we fix the input, **x**, into a network and ask how the output of the network, $f(\mathbf{w}, \mathbf{x})$, moves on the functional manifold, $\mathcal{F}$, given an infinitesimal weight perturbation, **du**, in $W$ where $\mathbf{w_d} = \mathbf{w_t} + \mathbf{du}$. For an infinitesimal perturbation **du**,

$$f(\mathbf{x}, \mathbf{w_t} + \mathbf{du}) \approx f(\mathbf{x}, \mathbf{w_t}) + \mathbf{J_{w_t}}\, \mathbf{du}, \tag{1}$$

where $\mathbf{J_{w_t}}$ is the Jacobian of $f(\mathbf{x}, \mathbf{w})$ for a fixed **x**, $J_{i,j} = \frac{\partial f_i}{\partial w^j}$, evaluated at $\mathbf{w_t}$. We measure the change in functional performance given **du** as the mean squared error

$$d(\mathbf{w_t}, \mathbf{w_d}) = |f(\mathbf{x}, \mathbf{w_t}) - f(\mathbf{x}, \mathbf{w_d})|^2 = \mathbf{du}^T\, (\mathbf{J_{w_t}}^T\, \mathbf{J_{w_t}})\, \mathbf{du} \tag{2}$$

$$= \mathbf{du}^T\, \mathbf{g_{w_t}}\, \mathbf{du}, \tag{3}$$

$$\tag{4}$$

where $\mathbf{g_{w_t}} = \mathbf{J_{w_t}}^T \mathbf{J_{w_t}}$ is the metric tensor evaluated at the point $\mathbf{w_t} \in W$. The metric tensor **g** is an $n \times n$ symmetric matrix that defines an inner product and local distance metric, $\langle \mathbf{du}, \mathbf{du} \rangle_{\mathbf{w}} = \mathbf{du^T}\, \mathbf{g_w}\, \mathbf{du}$, on the tangent space of the manifold, $T_w(W)$ at each $\mathbf{w} \in W$.

**Global paths in functional space:** Globally, we can use the metric to determine the functional performance change across a path connected set of networks. Mathematically, the metric changes as we move in $W$ due to the curvature of the ambient space that reflects changes in the vulnerability of a network to weight perturbation (Figure 1c). As a network moves along a path, $\gamma(t) \in W$ from a given trained network $\gamma(0) = \mathbf{w_t}$ to a sparse-hyperplane (hyperplane that consists neural networks of desired sparsity) such that $\gamma(1) = \mathbf{w_s}$, we can analyze the integrated impact of sparsification on network performance along $\gamma(t)$ by using the metric (**g**) to calculate the length of the path $\gamma(t)$ as:

$$L(\gamma) = \int_0^1 \langle \frac{d\gamma(t)}{dt}, \frac{d\gamma(t)}{dt} \rangle_{\gamma(t)}\, dt, \tag{5}$$

where $\langle \frac{d\gamma(t)}{dt}, \frac{d\gamma(t)}{dt} \rangle_{\gamma(t)} = \frac{d\gamma(t)}{dt}^T \mathbf{g}_{\gamma(t)} \frac{d\gamma(t)}{dt}$ is the infinitesimal functional change accrued while traversing path $\gamma(t) \in W$. As our objective is to find a path $\gamma(t) \in W$ that minimizes the total function length ($L(\gamma)$), we evaluate the geodesic between the dense network and its sparser counter part.

## 3    Geodesics

In this section, we evaluate the geodesic path, defined as the minimum functional path-length in weight space between any pair of networks. We also demonstrate that the formulation of high-performance paths holds the key to solving seemingly unrelated open-questions: (i) catastrophic forgetting and (ii) network sparsification.

**Network sparsification:** We pose the problem of finding a $p\%$ sparse counterpart ($w_s^p$) of a dense neural network ($w_t$) as finding a geodesic from ($w_{t1}$) to a $p\%$ sparse hyperplane ($W_s^p$).The sparse hyper-plane is the set of all networks that are $p\%$ sparse. As the geodesic minimizes the total distance traversed on the functional space, it ensures that the sparse network obtained will be both, functionally similar to the original network and high-performing.

**Catastrophic forgetting:** In order to obtain a single network well-trained on two sequential tasks, we train two networks on the two tasks independently ($w_{t1}$, $w_{t2}$) and pose the problem of finding a network that performs well on both the tasks, as a geodesic between the two trained networks ($w_{t1}$ and $w_{t2}$) on the functional space. The functional manifold used could pertain to either of the tasks (ensuring that a single dataset/task is used at a time).

**Geodesics mathematical machinery:** To find geodesics on $W$, we can solve the geodesic equation given by:

$$\frac{d^2 w^\eta}{dt^2} + \Gamma^\eta_{\mu\nu} \frac{dw^\mu}{dt} \frac{dw^\nu}{dt} = 0 \tag{6}$$

where, $w^j$ defines the j'th basis vector of the weights space $W$, $\Gamma^\eta_{\mu\nu}$ specifies the Christoffel symbols ($\Gamma^\eta_{\mu\nu} = \sum_r \frac{1}{2} g^{-1}_{\eta r}(\frac{\partial g_{r\mu}}{\partial x^\nu} + \frac{\partial g_{r\nu}}{\partial x^\mu} - \frac{\partial g_{\mu\nu}}{\partial x^r})$) on the manifold. The Christoffel symbols capture infinitesimal changes in the metric tensor ($\mathbf{g}$) along a set of directions in the manifold. They are computed by setting the covariant derivative of the metric tensor along a path specified by $\gamma(t)$ to zero.

We, specifically, compute geodesic paths, $\gamma(t)$, so that $\gamma(0) = \mathbf{w_{t1}}$ and $\gamma(1) \in W_s$ where $W_s$ is the sparsity hyper-plane (for network sparsification) and $\gamma(1) = w_{t2}$ (for the catastrophic forgetting problem). As the computation of the Christoffel symbols is both memory and computationally intensive, we propose an optimization algorithm to evaluate the 'approximate' geodesic in the manifold.

Given a trained network, our procedure updates the weights of the network to optimize performance given a direction of sparsification. To discover a geodesic path $\gamma(t)$, we begin at a trained network and iteratively solve for the tangent vector, $\theta(w)$, at every point, $\mathbf{w} = \gamma(t)$, along the path, starting from $\mathbf{w_{t1}}$ and terminating at the sparse hyperplane, $W_s^p$ (for sparsification) or network ($w_{t2}$) trained for task-2 (for catastrophic forgetting). We specifically solve

$$\text{argmin}_{\theta(\mathbf{w})} \langle \theta(\mathbf{w}), \theta(\mathbf{w}) \rangle_\mathbf{w} - \beta \, \theta(\mathbf{w})^T v_w \text{ subject to: } \theta(\mathbf{w})^T \theta(\mathbf{w}) \leq 0.01. \tag{7}$$

The tangent vector $\theta(\mathbf{w})$ is obtained by simultaneously optimizing two objective functions: (1) minimizing the increase in functional distance along the path measured by the metric tensor ($\mathbf{g_w}$) [min: ($\langle \theta(\mathbf{w}), \theta(\mathbf{w}) \rangle_\mathbf{w}) = (\theta(\mathbf{w})^T \mathbf{g_w} \theta(\mathbf{w}))$ ] and (2) maximizing the dot-product between the tangent vector and $v_\mathbf{w}$, vector pointing in the desired direction [max: $(\theta(\mathbf{w})^T v_\mathbf{w})$] to enable movement towards the sparse hyperplane or network $w_{t2}$.

Our optimization procedure is a quadratic program that trades off, through the hyper-parameter $\beta$, motion towards the damage hyper-plane and the maximization of the functional performance of the intermediate networks along the path (elaborated in the appendix). The strategy discovers multiple paths from the trained network $\mathbf{w_{t1}}$ to $W_s$, sparse hyper-plane, where networks maintain high functional performance during sparsification. Of the many paths obtained, we can select the path with the shortest total length (with respect to the metric $\mathbf{g}$) as the best approximation to the geodesic in the manifold.

**Geodesics for network sparsification:** We apply the geodesic strategy to discover high-performance paths from a trained network (VGG11 on CIFAR-10) to a sparse hyperplane (blue curve in figure-2A). Here, the sparse hyperplane is defined as all networks that have zeroed out 50 (out of 60) conv-filters from layer-1 in VGG11. We compared our strategy with conventional heuristic fine-tuning (figure-2A) and demonstrate that the geodesic procedure is both rationale and computationally efficient. Specifically, an iterative prune-train cycle achieved through structured pruning of a single node at a time, coupled with SGD re-training [6, 3] (Figure 4A) requires 70 training epochs to identify a sparsification path. However, our geodesic strategy finds paths that quantitatively out-perform the iterative prune-train procedure and obtains these paths with only 10 training epochs (figure 2A,B).

In figure-2C, we apply the geodesic strategy to find a path from a trained multi-layer perceptron (MLP) to its 70% sparser counterpart. The geodesic discovers a network performing at a test-accuracy of 96.8% on the 70% sparse hyperplane.

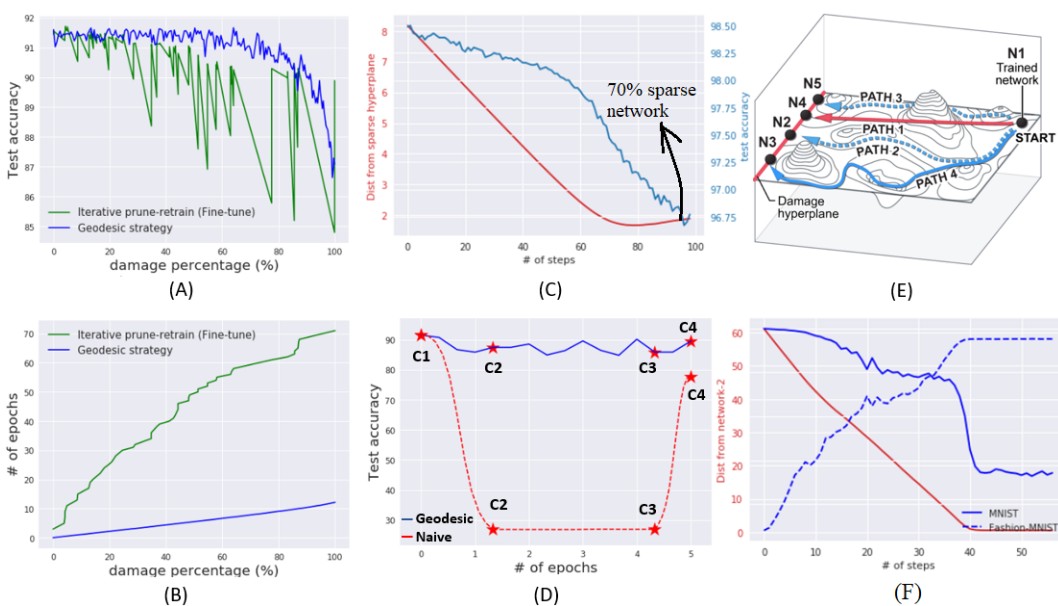

Figure 2: **Geodesic paths allow damage compensation through weight adjustment**: **(A)** Test accuracy (A) and **(B)** number of network update epochs (B) for geodesic recovery (blue) vs fine-tuning (green) while 50 (out of 60) conv-filters are deleted from layer1 in VGG11. Geodesic recovery requires ≤10 total update epochs. **(C)** Traversing the geodesic from an MLP (trained on MNIST) to its 70% sparser counterpart. The blue line captures the test-accuracy of the networks along the path and red-line captures the euclidean distance of the network from the sparse hyperplane ($W_s^p$). **(D)** Geodesic strategy (blue) allows networks to dynamically transition between configurations: C1, trained VGG11 network ; C2, 50 conv-filters removed from C1; C3, 1000 additional nodes removed from classifier-layers in C2; C4, 30 conv-filters in conv-layer1 restored to C3. Dynamic transitioning enabled within 5 epochs. Naive strategy is red). **(E)** A depiction of multiple sparsification paths on the loss landscape from trained network (N1) to networks on the sparse (damage) hyper-plane (N2, N3, N4, N5). The z-axes is network-loss, while the x,y axes are neural net weights. **(F)** Traversing the geodesic from network-1 trained on MNIST to network-2 trained on Fashion-MNIST along the MNIST functional manifold. The solid blue line is the test-accuracy of the networks (along the geodesic) on MNIST, while the dashed blue-line is the test-accuracy of the networks on Fashion-MNIST, and the red-line is the distance between from network-2. At the point of intersection of the solid and dashed blue line, we find networks that perform at 80% for both the tasks.

Additionally, the same geodesic strategy enables us to dynamically shift networks between different weight configurations (eg from a dense to sparse or vice-versa) while maintaining performance (Figure 2D). The rapid shifting of networks is relevant for networks on neuromorphic hardware to ensure that the real-time functionality of the hardware isn't compromised while transitioning between different power configurations.

**Geodesics for alleviating catastrophic forgetting:** In this section, we present our preliminary results on the geodesic framework formulated to alleviate catastrophic forgetting.

We chose two datasets (MNIST and Fashion MNIST) and train our network on both tasks one after the other. On doing so, we notice that the network initially performs at an accuracy of 99% on MNIST (after training on MNIST alone), but drops its performance to 10% after being trained with Fashion MNIST. This is a clear demonstration of the network catastrophically forgetting.

To alleviate this issue, we train 2 networks (on MNIST and Fashion MNIST). We find a geodesic path from $w_{t1}$ to $w_{t2}$ on the functional manifold (MNIST dataset). As we traverse the geodesic, we obtain a set of networks along the path that perform well on both MNIST and Fashion-MNIST (without having to be trained on both datasets simultaneously!). In figure-2F, we evaluated the geodesic path between the 2 trained networks and show that some networks along the geodesic have an accuracy of 80% on both the MNIST and Fashion MNIST dataset, without requiring rigorous training using both datasets simultaneously.

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
