# OpenReview forum: "Sparsifying networks by traversing Geodesics"
_NeurIPS.cc/2020/Workshop/DL-IG — NeurIPSW 2020: DL-IG Poster_

### Official Review · AnonReviewer2 · 2020-10-20
**Review of "Sparsifying networks by traversing Geodesics"**

**Rating:** 7
**Confidence:** 4

**Review:**

The authors interpret neural networks as functions parametrized by weights w. The goal is to alter the weights of the network so as to sparsify them, while simultaneously ensuring that the function parametrized by the original weights is close to the function parametrized by the sparsified weights. The authors interpret this as trying to find the optimal projection of the starting weights on the space of the desired (sparse) weights, where the distance is measured in terms of the change in the function value. The authors propose a heuristic algorithm to do the projection and demonstrate via experiments that the method works in practice.

I have two main comments:
(1)  I was confused by the terminology "length of the path" in equation (5). I believe to make it the length, the authors need to have a square root sign inside the integral.
(2) The explanation assumes that the input variable x is fixed. This means that the path obtained, say from w0 to w1, satisfies f(w0, x) \approx f(w1, x). But what about other values of x? Won't there be a different "optimal" path for each value? If so, how to pick the best path?

My suggestions would be: Please clarify the point (1) above. Will the implemented algorithm change if you require a square root sign? For point (2), perhaps the authors need to integrate out x and look at the expected path length?

---

### Official Review · AnonReviewer2 · 2020-10-30
**Geodesics in weight space**

**Rating:** 7
**Confidence:** 4

**Review:**

This work uses the weight jacobian of a neural network to define a local metric tensor.  With that metric tensor, geodesic lengths can be computed for any path through parameter space.  With a notion of distance in parameter space, geodesics can be defined (shortest length paths), and the paper investigates what happens when one tries to find the shortest length path between some initial configuration and some desired hyperplane, be it one that is sparse since we are interested in model compression, or a parameter settings that does well on a different task since we want to avoid catastrophic forgetting.  This requires solving a quadratic program that tries to dually minimize the path length while headed in a desired direction.

I like the overall idea quite a bit and the results look promising.  I have to admit I don't fully understand what has happened in the catastrophic forgetting case, what is the target in this case, is it the specific weight configuration found when trained on FashionMNIST?  How does this performance compare to a baseline linear interpolation between the two parameter values?

Also, overall why would we settle for a notion of distance that is inherently euclidean in the function outputs of the network? Why not instead use the Fisher information metric?  If I'm not mistaken this shouldn't be much more expensive, as for the typical cross entropy loss style function if we simply sandwich the hessian of the cross entropy loss with the jacobians that will give us the Fisher information metric.  How do paths that are geodesic in this metric compare to the ones euclidean in the output of the network?

Overall interesting work and intriguing results but it left me wanting a bit more.

---

### Decision · Program_Chairs · 2020-11-07

Accept (Poster)